# *ThermalWrist*: Smartphone Thermal Camera Correction Using a Wristband Sensor [note 1]

**DOI:** 10.3390/s19183826

**Published:** 2019-09-04

**Authors:** Hiroki Yoshikawa, Akira Uchiyama, Teruo Higashino

**Affiliations:** Graduate School of Information Science and Technology, Osaka University, Osaka 565-0871, Japan (A.U.) (T.H.)

**Keywords:** thermal camera, offset correction, skin temperature, smartphone, wristband sensor

## Abstract

Thermal images are widely used for various healthcare applications and advanced research. However, thermal images captured by smartphone thermal cameras are not accurate for monitoring human body temperature due to the small body that is vulnerable to temperature change. In this paper, we propose *ThermalWrist*, a dynamic offset correction method for thermal images captured by smartphone thermal cameras. We fully utilize the characteristic that is specific to thermal cameras: the relative temperatures in a single thermal image are highly reliable, although the absolute temperatures fluctuate frequently. To correct the offset error, *ThermalWrist* combines thermal images with a reliable absolute temperature obtained by a wristband sensor based on the above characteristic. The evaluation results in an indoor air-conditioned environment shows that the mean absolute error and the standard deviation of face temperature measurement error decrease by 49.4% and 64.9%, respectively. In addition, Pearson’s correlation coefficient increases by 112%, highlighting the effectiveness of *ThermalWrist*. We also investigate the limitation with respect to the ambient temperature where *ThermalWrist* works effectively. The result shows *ThermalWrist* works well in the normal office environment, which is 22.91 °C and above.

## 1. Introduction

Thermal cameras are widely used to monitor the temperature [1] because they can measure temperature distributions quickly without physical contact. For this reason, they are used in various fields such as animals [2,3], agriculture [4], buildings [5] and humans. For example, thermal cameras are applied to humans for anomaly detection such as heatstroke [6] and infection [7]. Human skin temperature is also used for estimating various psychological states such as stress and thermal comfort [8,9,10]. Although some wearable devices with skin temperature sensors are available (e.g., E4 wristband [11] and Microsoft band 2 [12]), their applications are still limited due to the limitation of single point measurement.

Recently, thermal cameras are widely used because low-cost and miniaturized models are available on the market. For example, FLIR ONE [13] is a low-cost thermal camera attached to a smartphone. Thanks to such smartphone thermal cameras, we can measure our skin temperature anytime anywhere. However, the accuracy of the low-cost thermal camera is insufficient to monitor the skin temperature compared with the high-end models.

Thermal cameras are categorized into two types: those with cooled infrared detectors and those with uncooled infrared detectors [14]. The performance of the cooled detectors is much higher than the other although they are bulky and expensive due to the cooling apparatus [15]. Therefore, the smartphone thermal cameras are with uncooled infrared detectors. The uncooled infrared detector element converts its temperature rise to electric signals. The temperature of the object principally calculated from the signals and its emissivity. The measured temperature greatly fluctuates in accordance with the parameters configured by the user, the camera’s body heat, the efficiency of the element, and the packaging method [14]. By these effects, the measurement error of the smartphone thermal camera is larger than the high end one. The error range of FLIR ONE is ±3 °C or ±5% which is larger than the range of human skin temperature changes in daily life. This is clearly not enough for various healthcare applications. For example, Abdelrahman et al. [8] reported that the nose temperature decreased by 1.49 °C when participants read a literary piece accompanied by increasing the cognitive load.

To overcome this problem, we propose *ThermalWrist*: a dynamic offset correction method for thermal images of faces captured by smartphone thermal cameras. We fully utilize the key feature of the thermal cameras: the measurement fluctuation is mainly caused by the offset which is common in all the pixels in a single thermal image [16]. In other words, we can measure the difference of temperature correctly between any pair of pixels in the same thermal image even using the smartphone thermal camera. There is a method that uses the temperature reference fixed in the environment. It is actually used in airports, etc. However, *ThermalWrist* does not require any fixed devices and achieves an accurate mobile thermal measurement. *ThermalWrist* combines thermal images with a reliable absolute temperature obtained by a wristband sensor based on the above feature. First, we obtain a thermal image including a reference point (a wrist or a palm) of which absolute temperature is measured by a wearable device (a wristband sensor). Second, we estimate the offset at the reference point by comparing the temperature measured by the thermal camera and the wristband sensor. Finally, *ThermalWrist* corrects temperature in the thermal image by adding the offset to all the pixels.

Since we assume measuring the skin temperature, the reference point should be any point, which has an emissivity as large as the face. We use the wristband sensor as an accurate thermometer, which is widely used in daily life. Since the wristband sensor covers the measurement point, the thermal camera cannot directly capture the temperature of the same point. Therefore, we define the reference point as the point that has a high correlation with the point measured by the wristband sensor. In this paper, we use a palm or a wrist for the reference point and compare the performance through the real experiment.

To evaluate *ThermalWrist*, we conducted real experiments with eight subjects in an indoor air-conditioned environment. The mean absolute error and the standard deviation of the face temperature measurement error decrease by 49.4% and 64.9%, respectively, and Pearson’s correlation coefficient increases by 112%, highlighting the effectiveness of *ThermalWrist*. In addition, we show that the combination of *ThermalWrist* and the correction by averaging continuous observations improve the evaluation values. Finally, in order to evaluate the influence on *ThermalWrist* due to environmental temperature change, we conducted an additional experiment in the cool environment and reveal the limitation with respect to the ambient temperature.

The main contributions of this paper are summarized as below:
We propose *ThermalWrist*: an offset correction method for thermal images of faces measured by a smartphone thermal camera by using a reference point combined with the specific feature of thermal cameras without any fixed devices.We compare the two reference points and discuss their features through the real data with 1148 samples collected from 11 subjects.We investigate the limitation of *ThermalWrist* on ambient temperature and show it can be used in the indoor air-conditioned environment.


## 2. Related Works

### 2.1. Applications Using Skin Temperature

Many research works have revealed the relation between human mental states and skin temperature. Choi et al. (2012) [17] investigate the possibility of the use of human body skin temperature to assess thermal sensation using the temperature sensor on the skin. Tag et al. (2017) [18] present the system for tracking positive cognitive and emotional states by using temperature sensors on eyeglasses. Genno et al. (1997) [19] use facial skin temperature to evaluate stress and fatigue. They revealed the fatigue is the load of the stress by assuming the estimation formula. Understanding the stress of humans is a crucial issue in our society. Recently, Japanese companies are obligated to conduct the stress check test for the employees by law [20]. Smartphone thermal cameras are one of the key technologies to record the stress levels ubiquitously, noninvasively and automatically.

### 2.2. Applications Using High-End Thermal Cameras

Thermal cameras have recently been used for the estimation of human thermal comfort. Burzo et al. (2014) [21] divide the thermal comfort into three levels: “hot discomfort”, “comfort” and “cold discomfort” and combine other biosensors with a thermal camera to estimate the thermal comfort. Ranjan et al. (2016) [9] estimate the thermal sense using thermal images and propose a method to reduce energy consumption in buildings. Stress or cognitive load estimation methods are shown by many research works [8,19,22,23,24,25,26]. Abdelrahman et al. (2017) [8] present an unobtrusive indicator of users cognitive load based on thermal images by monitoring forehead and nose temperature. The other emotions are estimated by monitoring the skin temperature [27]. Pavlidis et al. (2002) [28] detect lies based on bloodstream increase estimated by thermal images. López et al. (2015) [29] and Basu et al. (2015) [30] propose methods to estimate human emotion by facial temperature distribution. Thermal cameras are also useful to measure such psychological states since they do not disturb user behavior.

However, most of the existing works, shown in Table 1, use a high-end thermal camera that has high accuracy such as ±2 °C or ±2%, or higher. In reality, we can not always use such high-end thermal cameras, causing the problem of frequent fluctuations of measurement by low cost (i.e., smartphone) thermal cameras.

### 2.3. Application Using Smartphone Thermal Cameras

Some applications using smartphone thermal cameras have been developed since their release. Nurmi et al. (2017) [35] developed a low-cost solution for search and rescue operations using smartphone thermal cameras. Mauriello et al. (2017) [36] evaluated energy-efficiency issues in the building environment. However, they use smartphone thermal cameras for measuring objects whose temperature range is much wider than human skin temperature. In addition, there are some applications for medical support. For example, Jaspers et al. (2017) [37] use the smartphone thermal camera for the burn wound assessment. However, they measure temperature differences between the burn wound and healthy skin. As far as we know, there are no applications to measure human skin temperature using only smartphone thermal cameras. This is because smartphone thermal cameras are suitable for the measurement of relative temperature rather than absolute temperature.

### 2.4. Calibration of Thermal Cameras

To measure accurate temperatures, thermal cameras need to calibrate [16]. The purpose of calibration is to determine the accurate quantitative relations between camera output and incident radiation. Malmivirta et al. (2018) [38] mitigate the smartphone thermal camera’s error using a deep learning-based calibration technique. They achieve smartphone thermal camera calibration without any additional device; however, an error larger than 1 °C still remains. Our target scenarios are measurement of human skin temperature change over time, which requires higher accuracy. The calibration procedure introduced by Vollmer et al. (2011) [16] uses a black body with different temperatures whose emissivity is close to unity. In addition, their radiometric quantities and spectral quantities are well defined. Therefore, we obtain relations between the black body temperature and the temperature of the black body in the thermal image. Instead of the blackbody, shutter-based calibration [39] is widely used since a blackbody is large, heavy, and expensive. However, it is effective only when the shutter’s temperature is stable. Because of the small body of a low cost thermal camera, the shutter temperature tends to increase due to the increase of the internal temperature. Our method is inspired by the object-based calibration and leverages different parts of the same body with similar emissivity for calibration.

Several factors causing errors still exist in the thermal camera in a real environment. The emissivity of a target object is one of the factors, which is the efficiency of the surface in thermal energy emission. The emissivity is a specific parameter dependent on a component of substance. Mitchell et al. (1967) [40] reported the difference of the emissivity of human skin is 0.95–0.99, caused by the difference of the blood flow and skin color. This 4% difference leads to 0.25 °C error in the measurement of skin temperature. In order to determine the emissivity, it is necessary to measure the temperature of an object whose emissivity is known, such as black body tape [41]. To solve the problem, we use the body parts of the same person as the reference and the target. Therefore, *ThermalWrist* is not affected by skin color and blood flow because the emissivity of the reference and the target is almost the same. We note that our target environment is daily life where blood flow does not change greatly. However, the correction of the other factors is still challenging. Electrical noise and noise due to fluctuation in the scaling of thermal energy are the remaining major factors. To mitigate these effects, we essentially need high-end thermal cameras if we do not rely on any additional devices. Our method tackles this challenge by a combination of a wristband sensor and a smartphone thermal camera.

## 3. *ThermalWrist*: Smartphone Thermal Camera Correction Method

### 3.1. Overview

Infrared detectors built in thermal cameras are composed of a number of detector elements having different signal responsibilities. To measure quantitatively, the nonuniformity is corrected using a method called nonuniformity correction (NUC) [16]. Because of NUC, there is little difference in the accuracy between detector elements. Therefore, the error of a thermal image is a common offset over all pixels.

We correct the offset by setting a reference point in a thermal image. Figure 1 shows an overview of *ThermalWrist*. We assume that a user wears a wristband sensor such as an E4 wristband sensor to measure her/his wrist skin temperature in real time. The input is wrist temperature measured by the wristband, a visible image, and a thermal image. We extract the skin temperature of body parts from the thermal image by using image processing techniques. To do this, we apply pre-processing to match each pixel in the visible image with the temperature in the thermal image. Based on linear regression, the reference point temperature is estimated from the temperature of the wrist. By comparing the reference point temperature measured by the wristband sensor and the thermal image, we estimate the offset. Finally, we obtain the target point temperature by adding the offset to the thermal image.

### 3.2. Image Pre-Processing

As shown in Figure 2, the visible image and the thermal image have slightly different views due to the difference between camera positions. This means, for example, the palm pixels in the visible image are not exactly equivalent to those in the thermal image. Therefore, if we recognize a target region (e.g., a palm) in the visible image and then obtain the temperature distribution of the area from the thermal image, the distribution incorrectly contains temperatures of different parts. Figure 3 shows that this problem greatly affects temperature extraction, especially for a small part such as a fingertip and a nose. In addition, the position difference has a large impact on feature extraction from the distribution such as the minimum value and the maximum value. Therefore, we apply image pre-processing to match each pixel in the visible image with its correct temperature in the thermal image.

Our key idea in the pre-processing is that both images share almost the same edges. Figure 4 shows the detailed steps. First, edge detection using the Canny algorithm [42] is performed on both images to detect contours of body parts such as a face and hands. Then, smoothing based on a Gaussian filter is applied to both images to exclude noises after edge detection. Finally, Normalized Cross Correlation [43] (NCC) is used for template matching. NCC is used as a metric for evaluating the degree of similarity between a pair of images. In the template matching, we resize and slide the visible image over the thermal image to find the best position and size of the visible image that achieves the highest NCC with the thermal image based on the detected edges.

### 3.3. Reference Point Temperature Estimation

As we mentioned earlier, thermal cameras cannot capture the temperature of the point measured by the wristband sensor since it is covered by the wristband sensor itself. Therefore, we need to correlate the wristband temperature with the temperature of the reference points. For this purpose, we construct regression models given as
(1)$$\begin{array}{c} y=a+bx, \end{array}$$
where *x* is an explanatory variable which is a temperature reported by the wristband sensor, *y* is a target variable as a temperature of the reference point ($T_{\mathrm{est}}$) which is a palm ($T_{\mathrm{palm\_est}}$) or a wrist ($T_{\mathrm{wrist\_est}}$). *a* and *b* are parameters determined by training data. The appropriateness of the regression model is evaluated in Section 4.3. In addition, we evaluate the effect of the reference point selection in Section 4.4.

### 3.4. Reference Point Temperature Extraction

The temperature of any pixel in the visible image can be extracted from the thermal image after the pre-processing. In this paper, we propose two reference points close to the wristband. One is a palm and the other is a wrist around the wristband. In the following sections, we describe the temperature extraction for each reference point.

#### 3.4.1. Palm Temperature Extraction

We extract a palm from a visible image based on skin color as shown in Figure 5 because it is difficult to detect hands from thermal images. This is because the skin is easily cooled by the atmosphere in the cold environment and it is assimilated into the background in the thermal image.

We use an object detector using OpenCV based on a Haar-like feature [44] to detect a palm from a visible image. The position of the palm can be obtained as a rectangle. Since we can obtain the area including other parts such as a background and clothing, we need to further extract the palm region. For this purpose, we use an approach for human skin detection proposed by Tan et al. (2012) [45]. We use the green value of RGB color space and the saturation value of HSV (hue, saturation, value) color space [46] to obtain a squared histogram of color values in the face. We regard a pixel whose distance from the center of the smoothed histogram [47] is within three times of the standard deviation as a skin pixel. We extract the temperature of the skin pixel and calculate the average of the temperatures as a palm temperature $T_{\mathrm{palm\_meas}}$. The definition is given as
(2)$$\begin{array}{cc} T_{\mathrm{palm\_meas}} & =\frac{\sum_{(x,y)\in P}T\left(x,y\right)}{\left|P\right|}, \end{array}$$
where (x,y) is a coordinate in a visible image and *P* is a set of the extracted skin pixels. In addition, T(x,y) is the temperature of (x,y) and |X| is the number of elements in the set *X*. *P* is defined as
(3)$$\begin{array}{cc} P & =\{\left(x,y\right)|\left(x,y\right)\in R_{\mathrm{hand}}\cap C_{\mathrm{hand}}\}, \end{array}$$
where $R_{\mathrm{hand}}$ is a set of coordinates in the detected palm rectangle and $C_{\mathrm{hand}}$ is a set of coordinates whose colors are regarded as skin.

#### 3.4.2. Wrist Temperature Extraction

We extract a wrist from a visible image based on the wristband color. Different from the palm detector implemented in OpenCV, we need to implement the wristband detector. In this paper, we manually extracted rough positions of the wristband. However, we note that this may be easily achieved by attaching a special marker or collecting training data. The flow of the wrist temperature extraction is shown in Figure 6. First, to highlight the wristband, we conduct threshold processing based on the hue value from HSV color space. Next, we conduct edge detection, enclose the detected edges in a rectangle, and extract temperatures in the enclosed box from corresponding pixels of thermal images. Finally, we extract wrist temperature $T_{\mathrm{wrist\_meas}}$ by calculating the average of top 20% in the temperature distribution. $T_{\mathrm{wrist\_meas}}$ is defined as
(4)$$\begin{array}{cc} T_{\mathrm{wrist\_meas}} & =\frac{\sum_{(x,y)\in W}T\left(x,y\right)}{\left|W\right|}, \end{array}$$
where *W* is a set of the wrist coordinates that is defined as
(5)$$\begin{array}{ccc} W & = & \{\left(x,y\right)|\left(x,y\right)\in R_{\mathrm{wrist}}\cap C_{\mathrm{wrist}}\}. \end{array}$$


In the above equation, $R_{\mathrm{wrist}}$ is a set of coordinates in the enclosed box of the wristband and $C_{\mathrm{wrist}}$ is a set of coordinates whose temperature is in the top 20%, which achieves the highest r-squared value of 0.91.

### 3.5. Target Part Extraction

In this paper, we select faces as the target part because the face temperature is used in many applications [8,9,18,19]. We note that *ThermalWrist* can be also applied to other target parts. We use the face detector of OpenCV based on a Haar-like feature [44]. Since we can obtain a face position as a rectangle including a background, we need to extract the temperatures of facial skins from the thermal image. For this purpose, we use an approach for human skin detection [45] to the visible image and extract from the corresponding point in the thermal image. Figure 7 shows an example of skin detection. We extract face temperature $T_{\mathrm{face}}$ by calculating the average of face pixels. The definition is given by
(6)$$\begin{array}{ccc} T_{\mathrm{face}} & = & \frac{\sum_{(x,y)\in F}T\left(x,y\right)}{\left|F\right|}, \end{array}$$
where *F* is a set of coordinates in the detected face rectangle.

### 3.6. Offset Correction

We obtain the corrected temperature $T_{\mathrm{corr}}$ of the target by adding the offset *C* to the temperature $T_{\mathrm{target}}$ of the target in the thermal image as below:
(7)$$\begin{array}{cc} T_{\mathrm{corr}} & =T_{\mathrm{target}}+C,\\ C & =T_{\mathrm{est}}-T_{\mathrm{meas}}. \end{array}$$
The offset *C* is calculated for the reference point (either a palm or a wrist) by subtracting the temperature $T_{\mathrm{meas}}$ measured by the thermal camera from the temperature $T_{\mathrm{est}}$ estimated by the regression. Namely, $T_{\mathrm{est}}$ is either $T_{\mathrm{palm\_est}}$ or $T_{\mathrm{wrist\_est}}$ and $T_{\mathrm{meas}}$ is either $T_{\mathrm{palm\_meas}}$ or $T_{\mathrm{wrist\_meas}}$.

## 4. Evaluation

### 4.1. Evaluation Settings

For evaluation, we used FLIR ONE 2 and E4 wristband as a smartphone thermal camera and a wristband sensor, respectively. We also used a high-end thermal camera, FLIR T540, for the ground truth. The specifications of FLIR T540 and FLIR ONE 2 are shown in Table 2. The smartphone thermography has lower resolution and accuracy than the high-end one, but both of them have small thermal sensitivity and thermal differences can be accurately measured. E4 wristband measures the temperature at the sampling rate of 4 Hz. We use the mean temperature per minute for the evaluation. The accuracy of the temperature sensor is 0.2 °C within 36–39 °C and its thermal sensitivity is 0.02°C. It is accurate enough to calculate a reference point temperature.

To evaluate *ThermalWrist* in detail, we conducted two experiments in different ambient temperatures. All subjects gave their informed consent for inclusion before they participated in the study. The study was conducted in accordance with the Declaration of Helsinki, and the protocol was approved by the Ethics Committee of Graduate School of Information Science and Technology, Osaka University (201811). First, we collected the real data from 11 subjects (eight males and three females) in their twenties for a day. Each subject participated in the data collection at least four hours. In the experiment, we captured both visible and thermal images seven times with a ten-second interval every 30 min. During the experiment, the subjects were asked to wear E4 wristbands and worked as usual in the laboratory. They sat in front of the thermal cameras with the palms of both hands facing toward the cameras without any overlap. In total, 1148 samples (i.e., pairs of the visible and thermal images) were collected. The maximum and the minimum numbers of samples per subject are 105 and 63, respectively. The average room temperature of the experiment was 25.92 °C and the standard deviation was 0.69 °C. The highest and the lowest air temperatures in the room were 28.7°C and 23.4 °C, respectively.

In addition, we conducted an experiment in a roughly controlled cool environment to evaluate the influence on *ThermalWrist* due to environmental temperature change. We used the same equipment as the previous experiment and collected data from eight subjects in their early twenties for seven hours for each subject. Both of them participated in the previous experiment. We capture the images seven times with a ten-second interval every 30 min. In total, we collected 1085 samples. The average temperature of the room was 18.93 °C and standard deviation was 3.02 °C.

Hereafter, we denote the former experiment as “lab” and the latter experiment as “cool”. We note that the room temperature of the lab experiment is clearly warmer than the cool experiment.

### 4.2. Confirmation of a Thermal Camera’s Feature

We found that thermal cameras are suitable for measuring the difference of temperature in the same image even if a low-cost smartphone thermal camera. To confirm it, we compare the temperature difference between face and around wristband measured by the high-end thermal camera with the same measurement by the smartphone thermal camera. Figure 8 illustrates the relation between them in the lab experiment showing high correlation coefficient was 0.94. However, Figure 9 shows low correlation between the measurement by the high-end thermal camera and the smartphone thermal camera. The correlation coefficient was 0.21, which is totally different from the temperature difference. From the results, we see that smartphone thermal cameras can still capture temperature difference, which is comparable with the high-end one. We also note that temperature difference measured by the smartphone thermal camera tends to be slightly higher than the high-end one. This is because the difference of the resolution between the two thermal cameras. The temperature of each part from a thermal camera is calculated as a representative value (e.g., a mean or a maximum) in a particular area (e.g., a square or a circle), which is determined by each system. Therefore, the temperature measured by thermal cameras are affected by the temperature distribution which can be captured by each thermal camera. In particular, a temperature measured by a thermal camera with high resolution tends to be low in a cool environment. The evaluation results are shown in Section 4.6.

### 4.3. Reference Point Temperature Estimation

We evaluated the performance of the temperature estimation of the reference points by linear regression. The purpose of the experiments is to build models to estimate the reference point temperature from the wristband temperature. For this purpose, we use thermal images after the outlier removal to collect the training data. The training data can be collected in the continuous measurement as described in Section 4.7.

The regression functions are sufficient to understand the temperature difference between palm/wrist and wristband. The details are described in Appendix A. The regression function for estimating the palm temperature is given as:
(8)$$\begin{array}{cc} T_{\mathrm{palm\_est}} & =-3.4223+1.0639\;T_{\mathrm{wristband}}. \end{array}$$


The mean absolute error was 0.88 °C.

On the other hand, the regression function for estimating the wrist temperature is given as:
(9)$$\begin{array}{cc} T_{\mathrm{wrist\_est}} & =3.5528+0.8748\;T_{\mathrm{wristband}}. \end{array}$$


The mean absolute error was 0.79 °C.

From the above results, we see that the wrist temperature can be estimated more accurately than the palm. This is natural because of the closeness to the point measured by the wristband sensor. However, the errors of the two methods are much smaller than the error of the smartphone thermal camera; both are valid as reference points. We further investigate the performance of *ThermalWrist* based on the above regression functions in the next section.

### 4.4. Dynamic Offset Correction

First, we evaluate our method in the air-conditioned lab environment. Figure 10 shows the error distributions of the baseline (without correction), our palm-referenced method, and our wrist-referenced method. Palm (proposed) and Wrist (proposed) are the proposed methods including the reference point temperature estimation. On the other hand, Palm (ideal) and Wrist (ideal) are the ideal methods that use the reference point temperatures captured by the high-end thermal camera instead of the estimated temperatures. Therefore, the ideal methods are more accurate than the proposed methods because they are not affected by the accuracy of the reference point temperature estimation. We see that the distribution approaches 0 and the error dispersion becomes smaller in the proposed methods. Especially, the proposed method succeeded in completely excluding the large error in the baseline. Although such large error might be removed by outlier detection using multiple images captured over a long time (e.g., one minute), the advantage of our method is the dynamic offset correction with a *single image*. We also found the error distributions slightly shift to positive. One of the reasons is the fact that the average temperature in a small region (e.g., a wrist) tends to become higher than that of a large region (e.g., a face) when the resolution is high.

To investigate the statistical significance of *ThermalWrist*, we conduct the Brunner–Munzel test [48] between the absolute error of the baseline and the absolute error of each method because the error of the baseline is not normally distributed. It can analyze equality between two independent samples with not assuming the normal distribution. Both of the *p*-values between the baseline and Palm (proposed) and between the baseline and Wrist (proposed) are much smaller than 0.001. This result means significant improvement of *ThermalWrist* compared to the baseline.

In addition, Figure 11a–c show the mean absolute error (MAE), the standard deviation of the error (SD), and the Pearson’s linear correlation coefficient (CORR) of the baseline and our methods, respectively. It is obvious that both of our methods reduce MAE and SD compared to the baseline. We also see that the Wrist (proposed) method and Wrist (ideal) method achieve smaller SD than Palm (proposed) and Palm (ideal), respectively. This is because the accuracy of the reference point temperature estimation for the wrist is higher than the palm as mentioned in Section 4.3. Finally, the CORR of our methods is remarkably higher than the baseline. This means that our methods achieve higher linearity, which is generally important in correction of sensors.

From the above results, we have confirmed that the wrist-referenced method is better than the palm-referenced method. Therefore, we use the wrist-referenced method in the following evaluation.

### 4.5. Evaluation for Each Subject

We evaluate *ThermalWrist* for each subject in a lab environment using the wrist-referenced method. Figure 12a,b show the difference of the wrist-referenced method and the baseline in MAE and SD for each subject, respectively. In both figures, a positive value shows a reduction of MAE or SD by our method. The subjects A–I are male while the other subjects J–L are female.

In Figure 12a, we can see improvements in MAE for all subjects. Therefore, *ThermalWrist* can improve accuracy for the purpose of the absolute temperature comparison among several persons. In Figure 12b, we can see clear improvements in SD over all subjects except J. However, SD of J even without *ThermalWrist* is 0.37, which is quite small compared to most of the other subjects. This means that *ThermalWrist* can improve precision regardless of individual differences including gender impact. Therefore, *ThermalWrist* can measure temperature difference in the same person (e.g., a temporal change) accurately. In practical situations, skin temperature is used to estimate the mental state of the person. For this purpose, the skin temperature change (i.e., mental state change) of the same person is more important than the comparison of the absolute temperature among different persons. In this perspective, improvement in SD is more important than MAE because the accurate mental state estimation requires the individual training data. For the above reason, *ThermalWrist* is appropriate for such situations containing the individual training step.

### 4.6. Effect of Ambient Air Temperature

The above experiments have shown the effect of the reference point temperature difference on the performance. Table 3 shows the SD of the correction error in each environment. The SD is 0.38 °C larger than the lab environment.

Figure 13 shows the relation between the number of data and the error of *ThermalWrist*. The data in the dataset used for calculating the SD are extracted by setting the lower limit of the temperature measured by the wristband. When the cool data are mixed at 32.40 °C, the SD starts to increase. The wrist temperature border is shown in Figure 13 as the blue line. The mean room temperature when the border temperature was recorded was 22.91 °C. For this reason, we consider that the wrist temperature needs to be 32.40 °C or higher in order to utilize *ThermalWrist* with a small error.

### 4.7. Comparison with Continuous Measurement Correction

The accuracy of smartphone thermal cameras may be mitigated by conducting multiple measurements to remove outliers. To see the effect of such simple outlier detection, we captured both visible and thermal images seven times with a ten-second interval every 30 min. Then, we remove outliers from continuous measurement by defining a suitable threshold. Figure 14 shows an example of the outlier removal process. The black line is the mean of the seven measurements and the red lines are the thresholds. The thresholds are defined as the temperatures that are higher or lower than the SD of the seven continuous measurements.

Table 4 shows that MAE, SD, and CORR for the dataset where the above outlier removal was applied. We see that almost all the results are improved by the outlier removal. The MAE of the baseline is especially improved by the outlier removal, which is slightly better than *ThermalWrist*. However, the SD and CORR of the wrist-referenced method are much better than the baseline even with the outlier removal. This result indicates that *ThermalWrist* greatly improves the performance of smartphone thermal cameras, especially in terms of the ability to follow the changes of the ground truth. Therefore, we have confirmed that *ThermalWrist* is also useful for continuous measurement. We note that the baseline with outlier removal may be acceptable depending on situations since it does not require wristbands. Nevertheless, to apply outlier removal, users have to remain stable in front of thermal cameras. In contrast, *ThermalWrist* is able to correct a single thermal image in combination with a wristband, which is beneficial for users.

## 5. Conclusions

In this paper, we presented dynamic offset correction for a smartphone thermal camera using a wristband sensor for low cost and accurate temperature monitoring. The design of *ThermalWrist* is based on the key feature that the measurement fluctuation of the thermal cameras is due to the offset, which is common in all the pixels in a single thermal image. *ThermalWrist* estimates the temperature of the reference point by regression from the wristband temperature measurement. We selected a palm and a wrist for the reference points for comparison. Through the real experiment with 1148 samples from 11 subjects, we confirmed that *ThermalWrist* remarkably improves the accuracy on three evaluation metrics: the mean absolute error, the standard deviation of the error, and the Pearson’s linear correlation coefficient. The limitation of *ThermalWrist* depends on the variance of reference point temperature distribution. The experiment in a cool environment shows that the variance is larger when the temperature measured by a wristband sensor is lower. The wrist temperature needs to be 32.40 °C or higher in order to fully utilize *ThermalWrist*. This result indicates that *ThermalWrist* is effective in normal indoor ambient temperature. We note that the limitation of wristband thermometers is low reliability when people sweat heavily (e.g., vigorous exercise, walking in a hot environment). However, our target applications include comfort level estimation for air conditioning [9] and cognitive estimation in daily life [8]. We assume that most of these applications are used in air-conditioned indoor environments where people seldom sweat. Therefore, we are able to rely on a wristband thermometer as a reference. Our future work is to design a method that does not depend on ambient temperature and to correctly extract the temperature of the reference point from thermal images using other wearable sensors or other methods.

## Figures and Tables

**Figure 1 sensors-19-03826-f001:**
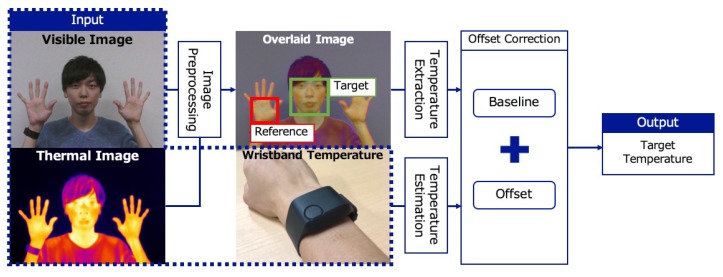
Overview of the offset correction.

**Figure 2 sensors-19-03826-f002:**
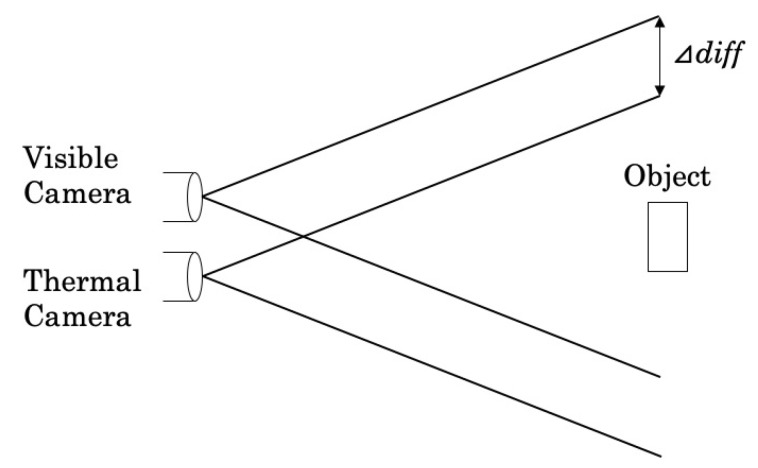
Position difference of visible and thermal cameras.

**Figure 3 sensors-19-03826-f003:**
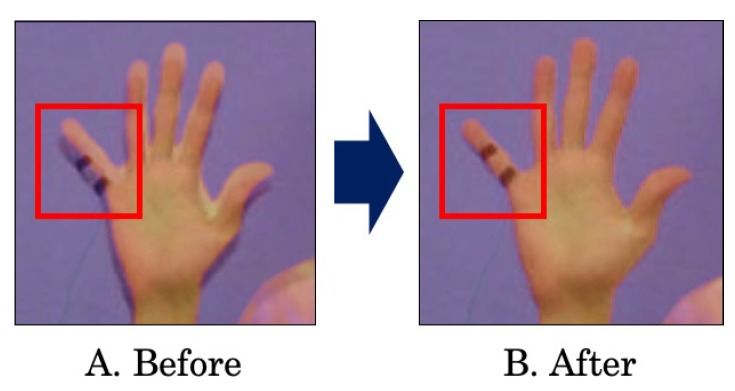
Result of overlapping.

**Figure 4 sensors-19-03826-f004:**
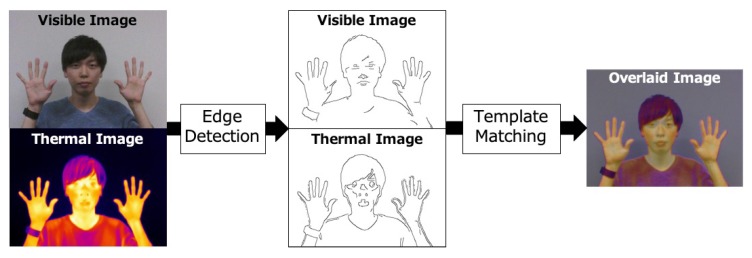
Image pre-processing steps.

**Figure 5 sensors-19-03826-f005:**
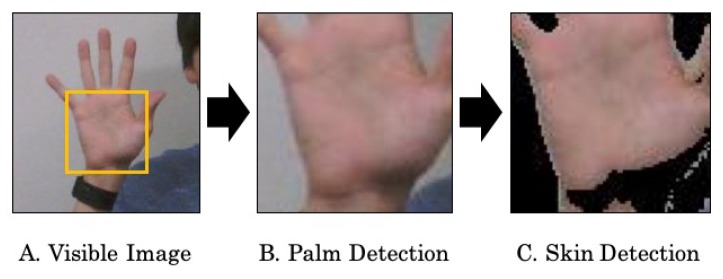
Palm region extraction.

**Figure 6 sensors-19-03826-f006:**
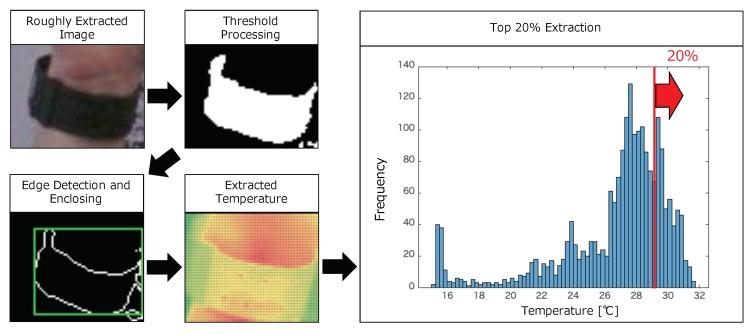
Flow of wrist temperature extraction.

**Figure 7 sensors-19-03826-f007:**
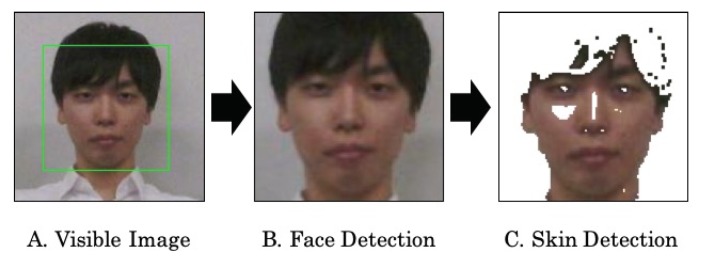
Face skin detection.

**Figure 8 sensors-19-03826-f008:**
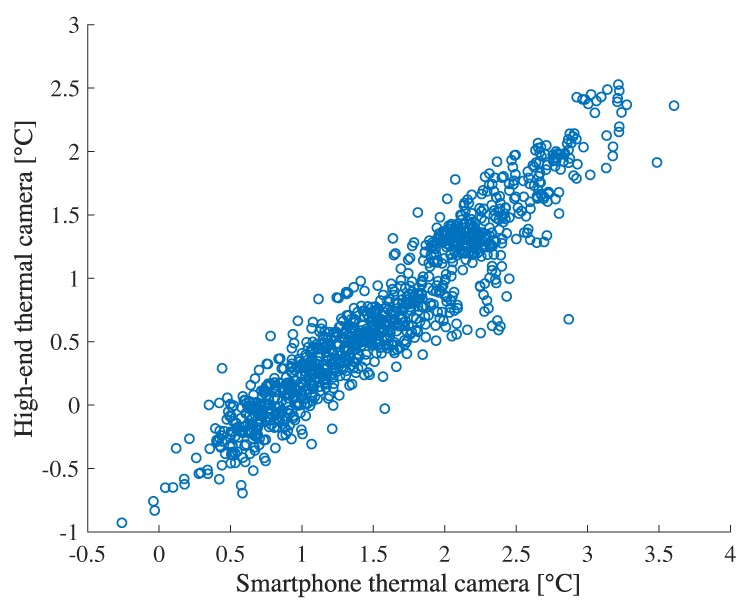
Relation between temperature differences (face–wrist) measured by a high-end thermal camera and smartphone thermal camera.

**Figure 9 sensors-19-03826-f009:**
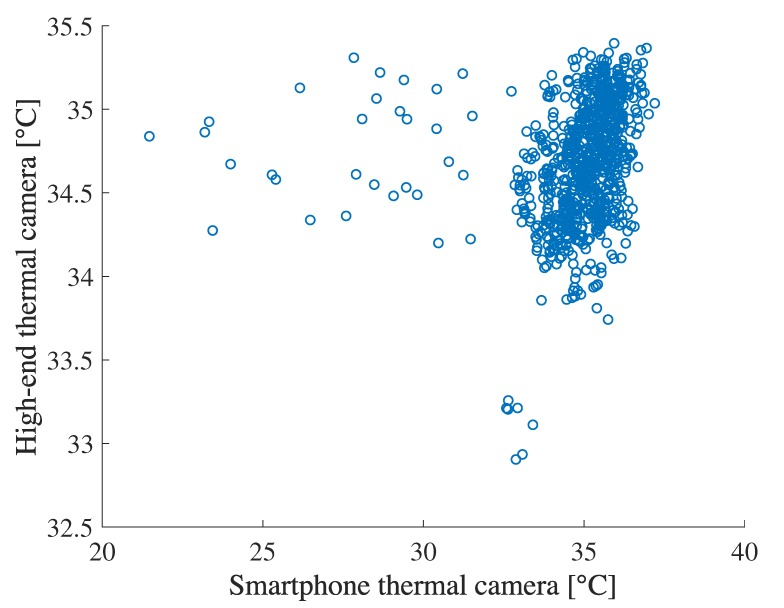
Relation between face temperatures measured by a high-end thermal camera and smartphone thermal camera.

**Figure 10 sensors-19-03826-f010:**
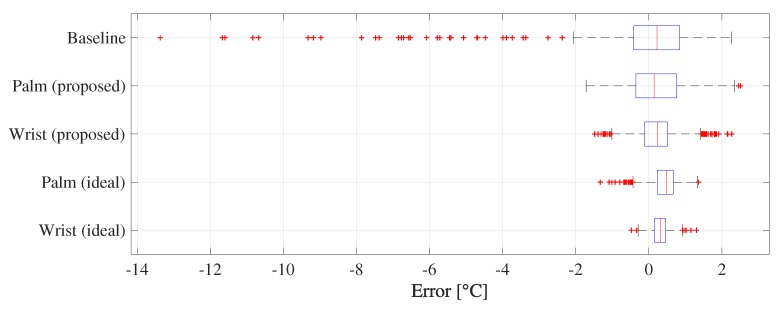
Error distributions of face temperature.

**Figure 11 sensors-19-03826-f011:**
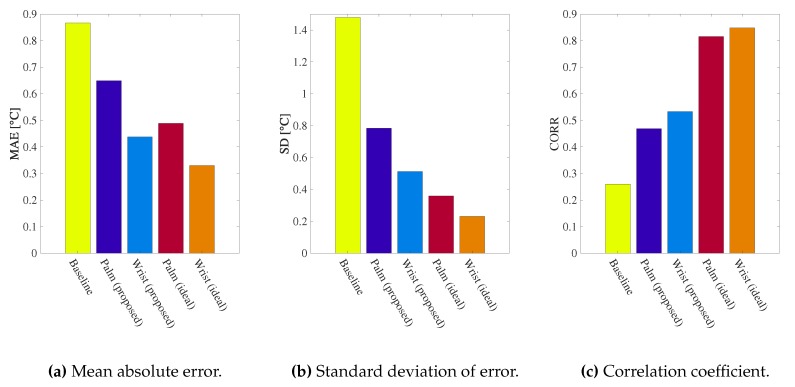
Evaluation in lab environment.

**Figure 12 sensors-19-03826-f012:**
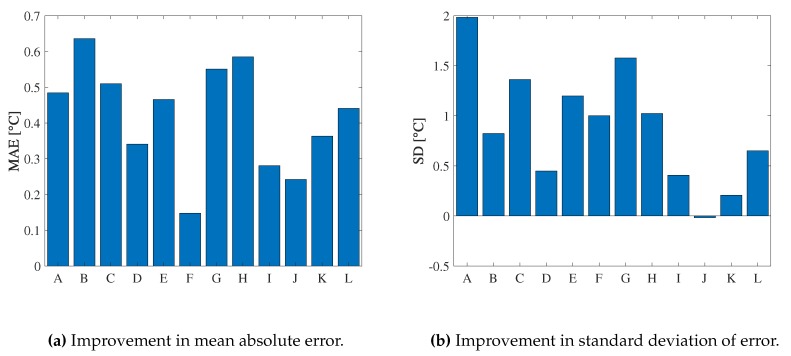
Improvement of *ThermalWrist* from baseline.

**Figure 13 sensors-19-03826-f013:**
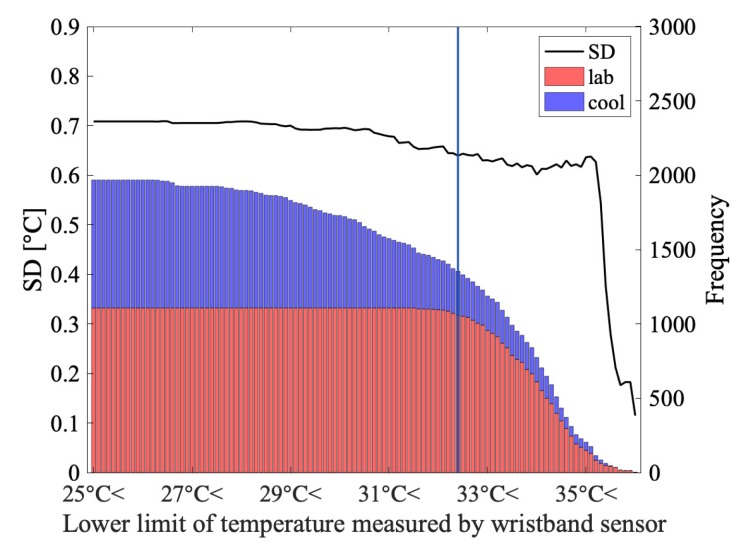
Relation between wristband measured temperature and SD of correction error.

**Figure 14 sensors-19-03826-f014:**
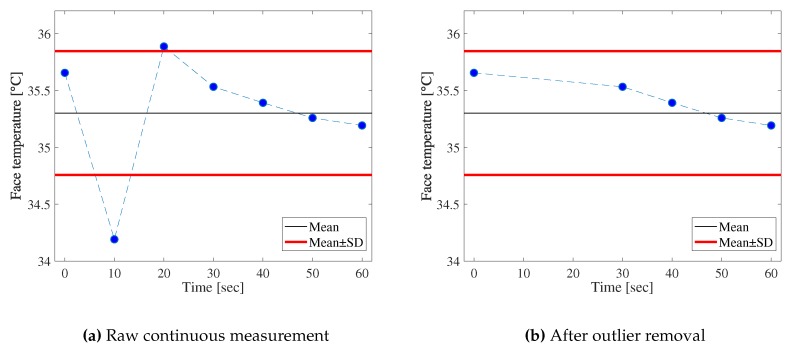
Example of removal defining thresholds.

**Table 1 sensors-19-03826-t001:** Thermal cameras used in existing works.

Author	Thermal Camera	Accuracy
Abdelrahman et al. (2017) [8]	Optris PI160	±2 °C or ±2%
Or et al. (2007) [23]	MikroScan 7200V	±2 °C or ±2%
Kang et al. (2006) [24]	MikroScan 7200V	±2 °C or ±2%
Shastri et al. (2012) [25]	FLIR SC6000	±2 °C or ±2%
Ranjan et al. (2016) [9]	FLIR A655sc	±2 °C or ±2%
Burzo et al. (2010) [21]	FLIR A40	±2 °C or ±2%
Basu et al. (2015) [30]	InfReC R300	±1 °C or ±1%
Gane et al. (2011) [31]	FLIR SC640	±2 °C or ±2%
Ebisch et al. (2012) [32]	FLIR SC3000	±1 °C or ±1%
Manini et al. (2013) [33]	FLIR SC660	±1 °C or ±1%
Hahn et al. (2012) [34]	testo 881	±2 °C or ±2%

**Table 2 sensors-19-03826-t002:** Specification of FLIR T540 and FLIR ONE.

	FLIR T540	FLIR ONE 2
Infrared Sensor Resolution	464 × 348	160 × 120
Accuracy	±1 °C or ±1%	±3 °C or ±5%
	(10–35 °C)	(0–35 °C)
Thermal Sensitivity	0.04 °C	0.15 °C

**Table 3 sensors-19-03826-t003:** Difference in standard deviation due to environmental change.

Method	Lab	Cool
Baseline	1.48	1.33
*ThermalWrist*	0.52	0.82

**Table 4 sensors-19-03826-t004:** Effect of outlier removal.

	MAE		SD		CORR	
Method	Before	After	Before	After	Before	After
Baseline	0.87	0.66	1.48	0.81	0.26	0.49
Palm-referenced	0.65	0.63	0.78	0.76	0.47	0.49
Wrist-referenced	0.44	0.43	0.52	0.46	0.55	0.57

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
