# Peer review of "ThermalWrist: Smartphone Thermal Camera Correction Using a Wristband Sensor †"

_sensors, 2019, doi:10.3390/s19183826_

Round 1
Reviewer 1 Report
This is the revised version. However, the authors have not addressed all reviewers' comments. In particular,
1. In section 3.2, the pre-processing of the thermal images has not clearly explained.
2. In section 3.3, the selection of reference point has not been clarified. The effect of the selection should be evaluated.
3. No comparisons with other works were made.
4. The modeling uncertainty has not been addressed.
Reviewer 2 Report
This paper proposed a novel method to mitigate errors in smartphone thermal cameras using the temperature of a reference point collected from a wristband type biosensor. Using smartphone thermal camera has a great potential to extend the applicability of thermal images. Also, integrating different sensors (i.e., camera and wristband) contributes to improving the performance of each sensor. As such, this study is worthwhile to study and relevant to the journal. However, there are several weak points that should be addressed before publishing the paper. Followings are the reviewer's comments and suggestions for improving the paper.
Major comments
1. The results are based on a very small number of subjects (8) calling into question the entire paper and its conclusion. Although there are numerous data collected from the subjects, the author is not sure about the generalizability of the findings from this study. Specifically, there are only two subjects and one subject for examining the effect of ambient air temperature and performance in continuous measuring. The limited number of individuals could create a strong dependency on the collected data point.
2. Also, the possibility of gender impacts should be noted since all subjects are male.
3. As discussed in the previous comment, this study analyzes multiple data points collected from subjects. This implies that the data has multiple levels (i.e., lower level - each data point, higher level - each subject). As such, statistical analyses in this study should consider the hierarchical structure of the data.
4. Line 370 ~ 380: It is difficult to argue that the performance of the proposed method is better than removing outliers in the baseline. Since removing outlier does not require wristbands, it might be better than the proposed methods. The authors should clearly argue the advantages of the proposed method.
5. The authors developed regression models based on the data collected from the high-end camera. However, the objective of this study is to mitigate errors in thermal images using smartphone thermal cameras and wristband. In this sense, the regression models should be developed based on the data collected from the smartphone thermal camera. The data collected from the high-end camera should be used only for evaluating the performance of the proposed method.
6. The reviewer strongly recommends including the results of statistical tests (e.g., r-square, p-value) in the paper.
Minor Comments
1. Several arguments in this paper should be supported by appropriate references. For example, statements on Line 32, Line 45, and Line 138 should be justified by references.
2. Line 65 – ‘effectiveness’ is not suitable in this context.
3. Line 123 – Ref(7) should be revised.
4. Line 141 – The statements mention the limitations of the proposed method. In the reviewer's opinion, it would be better to be located in the conclusion.
5. Several statements in this paper are red colored. The reviewer is not sure about the meaning of the red-colored statements.
6. Line 275 -
7. Table 3: The authors should clarify the meaning of each column. Does the second column mean the standard deviation of the errors? Also, the meaning of the correlation in the third column is not clear.
8. Line 376 – The authors provide appropriate references that support the outlier removal method used in this study.
Round 2
Reviewer 2 Report
The reviewer appreciates the authors' efforts to address the reviewer's comments. The comments were appropriately addressed.
This manuscript is a resubmission of an earlier submission. The following is a list of the peer review reports and author responses from that submission.
Round 1
Reviewer 1 Report
Fusion of data from different sources for the solution of a given task is the topic of your research. Methodology used is appropriate and confirmed with the reported experiments. Authors could consider reduction of the text, especially in the presentation of various experimental data, which could be provided as attachment if possible by the journal. Acceptance of publication in its form is suggested.
Author Response
We firstly express our appreciation for your helpful suggestions. We have reflected your comments by carefully considering them. Our responses to the comments, followed by a summary of the corresponding revisions, are given by the submitted file. Thank you in advance for your kind consideration of our manuscript.

Reviewer 2 Report
This paper works on the thermal camera correction. There are several points that need to be clarified:
1. Impact: The technology for thermal camera detection has been mature, so it is supposed that the method in this paper focuses on the thermal camera installed in the cellphones. In this case, it is not commonly acknowledged that thermal cameras are and will be widely adopted in the smartphones as there are lots of alternatives with much higher accuracy. Therefore, it requires more information to illustrate the significance of the proposed work.
2. Background: The error of the thermal cameras is concluded as 2 degrees or 2%. There are two key problems for this point, which need to be clarified to further enhance the impact of the paper. Firstly, most of the reference considered for this conclusion are published more than five years ago (i.e. earlier than 2014), which brings confusion whether they can still represent the technology up-to-date. Secondly, 2% is already a small percentage for error, and it remains few improvement space for future works.
3. Methodology: Thermal cameras and wrists with sensors are considered in the paper to work together. There is no sound technical proof and analysis for the correlation between them. As a result, it is hard to conclude the technical contribution because it seems like authors simply give a difference or a simple comparison between the data collected by the two.
In conclusion, the background should be reorganized to give a clear and advanced summary of the current development and limitation of thermal cameras. After that, the impact should be carefully addressed to make the paper referable and unique. Methodology should also be enhanced to provide scientific sound analysis.
Author Response
We firstly express our appreciation for your invaluable comments and helpful suggestions. By carefully considering each comment, we have addressed all the criteria. Our responses to the comments, followed by a summary of the corresponding revisions, are given by the submitted file. Thank you in advance for your kind consideration of our manuscript.

Reviewer 3 Report
p.p1 {margin: 0.0px 0.0px 0.0px 0.0px; font: 12.0px 'Helvetica Neue'; color: #454545} p.p2 {margin: 0.0px 0.0px 0.0px 0.0px; font: 12.0px 'Helvetica Neue'; color: #454545; min-height: 14.0px}=============
General comments
=============
This paper aims to correct miscalculated temperature from low cost thermal imaging based on the information additionally collected from a wrist biosensor. This is an interesting topic. And a wide range of techniques were implemented and explored. However, a current manuscript does not show strong contributions to this area given that its underlying principle is less convincing unfortunately, and authors need to properly discuss problems this paper tries to solve.
=============
Specific comments
=============
To help authors enhance this manuscript, I have made some comments below.
1. What is the main problem authors wants to solve?
While/After reading this manuscript, I have found it difficult to understand the main problem and the rationale for doing this work.
Page 2, line 43-44, authors mentioned that "the measurement fluctuation is mainly caused by the offset which is common in all the pixels in a single thermal image" and in line 50-51, "ThermalWrist corrects temperature in the thermal image by adding the offset to all the pixels" (using temperature measured by the wristband sensor)
The two sentences seem to be the key message from this paper. However, this underlying setup is the general setup for thermal imaging (especially, using high-end thermal cameras) just except for using the wristband - authors also cited Gane et al (2011) which describes this setup.
Authors also mentioned, page 4, line 116-118, "to mitigate these effects, we essentially need high-end thermal cameras if we do not rely on any additional devices. Our methods tackles with this challenge by a combination of a wristband sensor and a smartphone thermal camera" - not sure this proposed method tackles this issue as this method does not fundamentally improve the reliability of low-cost thermal camera.
In particular, authors argued that the accuracy of the low-cost thermal camera is insufficient to monitor the skin temperature compared with the high-end modes. I wonder if authors can provide the definition of accuracy. Generally speaking, thermal sensitivity is aligned with accuracy - and low cost thermal camera's sensitivity is very poor (>0.1K) in contrast with others (<0.01K) - this is the fundamental issue. Hence, it is very hard to say authors' method tackles this issue. Another issue is that this work is not considering with dynamic emissivity which is the key to consider in order to measure the accuracy of thermal measurement.
2. Literature Review
Authors have focused on low cost thermal imaging. However, fundamental issues are not properly addressed (in light with the comment above) and no literature on latest work using low cost thermal cameras is provided.
Please review advanced issues in thermal imaging - authors can easily find latest work by searching with some key words, for example, "low cost thermal imaging" or "mobile thermal imaging". At least, authors need to discuss those which use any of low-cost thermal cameras - no work is reviewed (Table 1).
and again, page 4, line 113-118, authors also highlighted the critical issue amid Emissivity. But this proposed work does not mitigate any issues of it. and high-end thermal camera also does not handle the issue. So algorithmic work should be further reviewed on this.
Some sentences do need references, eg page 3 line 81-84.
3. Technical issue
Another important point which I need to address as a reviewer is the thermometer in the Wristband.
The fundamental issue with the thermometer is its extremely low reliability. Empatica especially suffers with such issue, for example, when people sweats, the thermometer does not work properly (accuracy of this sensor is really low in this case). That's why researchers have generally avoided to use contact-based sensors in correcting measured temperature (ambient temperature sensors are required for the common setup for correcting).
Hence, the feasibility of this work in the real word situations sounds extremely low which is ironic because low cost thermal cameras are meant to be for real world situations. Otherwise, there is likely to be no need for low-cost solutions.
Sorry for criticising this work. But I sincerely hope my comments are helpful for your ongoing / upcoming research.
Lastly, thank you for all your effort!
Author Response
We appreciate the time and effort you have dedicated to providing insightful feedback on ways to strengthen our paper. Thus, it is with great pleasure that we resubmit our article for further consideration. We have incorporated revisions that reflect the detailed suggestions you have graciously provided. We also hope that our responses and the revisions we provide by the submitted file satisfactorily address all the issues and concerns the reviewer noted.

Round 2
Reviewer 2 Report
The manuscript presents a dynamic offset correction method for thermal images captured by smartphone thermal cameras. The topic of the paper is sound, and the flow is clear.
1.In section 3.2, the pre-processing of the thermal images has not clearly explained. For instance, the rationales of adopting Canny algorithm, Gaussian filter, and normalized cross correlation have not provided.
2.In section 3.3, the selection of reference point has not explained well. How would the position of the reference point affect the performance of the method?
3.It is suggested to evaluate the method with different reference points.
4.The method assumes that the relative temperatures in a single thermal image captured by smartphone thermal camera are highly reliable. Therefore, it is expected that the temperature differences measured by high-end thermal camera and by smartphone thermal camera should be the same. In Fig. 8, it is observed that the ranges of y-axis and x-axis are different (y=[-1.5,2.5] and x=[-0.5-3.5]) but the scatter plot in this figure is linear. It turns out that temperature differences measured by two cameras are not the same.
5.It is advised to compare with other existing schemes, to demonstrate the advantages of the proposed scheme.
Reviewer 3 Report
I appreciate authors for their effort in revising their manuscript.
However, this current manuscript still does not address the major problems I made earlier:
1) Novelty - the rationale for this work given the fact that this work is not novel in terms of its setup (using additional sensor to compensate for some accuracy issue of thermal imaging
- fundamentally, I couldn't find computational novelty from this work and this should be derived from the literature review.
2) literature review - fundamental issues are not properly addressed
- surprisingly, some papers are newly cited but some of them seem not to be relevant at all - e.g. [34] mainly uses depth cameras (Kinect etc) and its last study uses a FLIR thermal camera as just an infrared depth sensor - not for temperature monitoring. - I was wondering if authors properly reviewed the paper.
3) technical issues - low reliability of the thermometer
- authors mentioned "We note that the limitation of wristband thermometers is low reliability when people sweat. However, our target applications include comfort level estimation for air conditioning and psychological stress estimation in daily life. We assume most of these applications are used in air-conditioned indoor environment where people seldom sweat."
This is totally incorrect. Stress estimation is strongly related to sweat conditions. and authors mentioned their application in this area, but did not properly cite papers about low-cost thermal imaging for stress estimation or something at all.